# Risk and protective factors for incidents of intimate partner violence among active-duty military personnel

Valerie A. Stander[1]*, Travis N. Ray[1,2], Sabrina M. Richardson[1,2], Kelly A. Woodall[1,2], Cynthia J. Thomsen[1,2], Joel S. Milner[3], James E. McCarroll[4], David S. Riggs[5], Stephen J. Cozza[4]

**1** DoD Center for Deployment Health, Naval Health Research Center, San Diego, California, United States of America, **2** Leidos, Inc., San Diego, California, United States of America, **3** Center for the Study of Family Violence and Sexual Assault, Northern Illinois University, DeKalb, Illinois, United States of America, **4** Center for the Study of Traumatic Stress, Uniformed Services University of the Health Sciences, Bethesda, Maryland, United States of America, **5** Medical and Clinical Psychology, Uniformed Services University of the Health Sciences, Bethesda, Maryland, United States of America

* valerie.a.stander.civ@health.mil

## Abstract

### Purpose

Intimate partner violence (IPV) among military personnel harms service members and their partners and has implications for military readiness. Understanding modifiable risk and protective factors for IPV perpetration in this context is important for prevention and response.

### Method

Data from the Millennium Cohort Study (2011–2013) were used as a baseline in a hypothesized prospective path model predicting IPV perpetration. IPV outcomes were operationalized as reports to the Department of Defense Family Advocacy Program Central Registry that occurred after baseline and met Department of Defense criteria for psychological or physical domestic abuse.

### Results

A posttraumatic stress symptom cluster indicative of general negative affect and alcohol dependence mediated the effects of other posttraumatic stress symptoms—as well as the effects of protective factors (i.e., socioeconomic, psychosocial, physical health)—on risk of IPV perpetration. Only socioeconomic status had indirect, direct, and moderated effects on IPV perpetration.

**Data availability statement:** The data used in the analyses for this report are the property of the U.S. Federal government and are only available by permission due to government restrictions on their use. Specifically, they must be requested through with Defense Manpower Data Center (contact: https://dmdcrs.dmdc.osd.mil/dmdcrs/public/), as well as the Millennium Cohort Program Science Review Committee (contact: usn.nhrc-MillenniumCohortPI@health.mil).

**Funding:** Defense Health Agency, U.S. Department of Defense.

**Competing interests:** The authors have declared that no competing interests exist.

## Conclusions

Findings suggest future program development and evaluation should consider whether common protective factors—such as economic stability, better health (e.g., sleep quantity and quality), career satisfaction, and psychosocial factors (e.g., social support)—can be modified through integrated prevention to reduce risk for multiple interrelated outcomes (e.g., posttraumatic stress disorder, substance dependence, and IPV).

## Introduction

Intimate partner violence (IPV) is a significant public health problem with a range of exposures and impacts for victims. The Centers for Disease Control and Prevention identifies IPV as, "…physical violence, sexual violence, stalking and psychological aggression (including coercive tactics) by a current or former intimate partner (i.e., spouse, boyfriend/girlfriend, dating partner, or ongoing sexual partner)" [1,2]. Some potential consequences of IPV include increased risk for mental disorders such as depression, substance use, posttraumatic stress disorder (PTSD), and other anxiety-related conditions [3]. While IPV is a concern in all sectors of the U.S. population, there is some evidence that IPV prevalence rates may be elevated within the military [4], particularly for more severe forms of aggression [5,6], although more comparative empirical evidence is needed. Military service members are exposed to unique levels and types of stress that may increase risk, such as frequent moves that disrupt social support networks, combat deployment, and multi-trauma exposure. Operational stress, particularly deployment, may disproportionately increase risk for more severe IPV perpetration [7]. Also, aspects of military culture (e.g., hypermasculinity) and hierarchical structure may both increase risk of IPV and present challenges for help-seeking among junior ranking or civilian spouses experiencing it [8].

Involvement in IPV, whether as a perpetrator or victim, can negatively impact military personnel's well-being and compromise military readiness [9,10]. When service members are involved in IPV incidents, the Department of Defense (DoD) Family Advocacy Program (FAP) bears the cost and responsibility for evaluation and appropriate treatment [11]. When a service member is the perpetrator, there may be an independent legal investigation and punitive action under the Uniform Code of Military Justice and/or within civilian court systems, and cases have been on the rise [12]. Depending on the outcome of these processes, DoD may lose capable personnel. Also, under Public Law 104–208, Section 658, anyone convicted of assault or attempted assault on a family member is prohibited from handling firearms or ammunition, with no exceptions for military members. In light of such potential consequences, DoD maintains a registry documenting incidents that meet criteria for IPV as well as other forms of abuse or neglect in military families, and DoD policy requires ongoing efforts to analyze these data and identify factors that may improve prevention and treatment practices [13]. The objective of this study was to analyze data from the Millennium Cohort Study, integrating self-report data from the largest

longitudinal research cohort of military personnel to date with archival data from the DoD FAP Central Registry in order to better understand the interplay of putative risk and protective factors for IPV perpetration among service members.

## Research on risk factors

In a recent report using Millennium Cohort Study data [14], we explored multiple mental and behavioral health risk factors that are commonly elevated in military populations and that may be associated with IPV perpetration risk. Among a range of potential risk factors included in those analyses (e.g., posttraumatic stress symptoms [PTSS], depression, mild traumatic brain injury, and indicators of alcohol misuse or dependence), two factors most consistently emerged as predictors of IPV perpetration: PTSS and alcohol dependence. Both have been identified as significant risk factors for IPV perpetration in a number of other studies of military populations as well [10,15–18].

PTSD among military personnel is a particular concern for DoD during combat operations. PTSD diagnosis rates among deployed personnel have varied, but a meta-analysis of data from service members deployed to recent conflicts revealed rates ranging from 2% to 24% [19]. For the prior military conflicts in Iraq and Afghanistan (Operations Enduring and Iraqi Freedom), research suggests a prevalence of 12% overall and over 20% among those seeking care [20]. A review of the association between PTSD and IPV estimated true score correlations of.31 and.36 for physical and psychological IPV, respectively, and also indicated effects may be larger in military than in civilian populations [15].

Understanding how specific symptom patterns associated with PTSD may create vulnerabilities for outcomes such as IPV and how potential protective moderators may diminish those risks is important for both prevention and treatment. The current diagnostic criteria for PTSD include four symptom clusters [21]: (a) negative cognitions and mood (e.g., emotional numbing), (b) hyperarousal, (c) avoidance of trauma reminders, and (d) trauma re-experiencing (e.g., flashbacks, nightmares). Among these, there is substantial evidence that hyperarousal may be the symptom cluster most predictive of aggression, including IPV, and that this factor may mediate the impact of other PTSS clusters on perpetration risk [16,22–27]. Cognitive processing theories suggest hyperarousal may be uniquely associated with aggression because it inhibits the higher-level cognition necessary to interrupt automatic aggression in response to perceived threat [28,29]. Hyperarousal also can heighten anger, making aggression more likely [10,30–32]. Cognitive Action Theory, for example, suggests those with PTSD are stuck in survival mode, in which perceived threats to safety and threat-survival reactions are continuously primed [28,29]. This constant state of arousal may then become a mediator of the effect of PTSD on risk for aggression by interfering with cognitions that might lead to slower, more complex, and more adaptive reactions.

The significance of hyperarousal as a proximal predictor of IPV perpetration is potentially confounded by the fact that this cluster includes symptoms less specific to hyperarousal and more commonly overlapping with other comorbidities such as depression, or simply indicative of general negative affect (i.e., anger/irritability, sleep problems, and difficulty concentrating). Anger, for instance, is a significant predictor of aggression [14,31–35]. Our research and the work of others suggests that anger may mediate the association of hyperarousal with aggression [14,31,36,37]. However, sleep problems and difficulty concentrating also are likely to decrease complex cognition and increase the risk of aggression in reaction to perceived threat.

To address this possible confound, we previously suggested using a five-factor structure proposed by Elhai et al. [38] to model the influence of PTSD on IPV risk [14]. The Elhai model parses negative affect (i.e., anger/irritability, sleep problems, and difficulty concentrating) out of the hyperarousal cluster. In the current study, we extended our prior work using Elhai's five-factor model of PTSD to evaluate the separate roles of hyperarousal, negative affect, and other PTSS clusters—in conjunction with alcohol dependence—as potential risk factors for IPV perpetration.

## Research on protective factors

As in many areas of research, studies of IPV have focused more on risk than on protective factors [39]. However, Elbogen and colleagues [40,41] previously identified three types of factors that are protective against general aggression

among veterans: socioeconomic, psychosocial, and physical health. They found significant moderating effects. Specifically, among those at low cumulative risk, protective factors were minimally associated with aggression. However, among those at high risk, protective factors were more significantly associated with reductions in risk of self-reported aggression. Across categories, they identified several significant modifiable variables associated with reduced risk of aggression, including social support, perceived self-determination (i.e., control over one's life), economic stability, healthy sleep, and low levels of pain, and suggested that these should be targets for intervention through rehabilitative health services [40].

**Socioeconomic factors.** Members of the military are somewhat protected from economic stressors like joblessness, and they receive full healthcare coverage, housing allowances, reduced costs for childcare, and access to financial counseling, among other economic benefits [42]. Still, socioeconomic status is salient in the military because it is structured explicitly through enlisted versus officer status and pay grade. These status markers have been associated with IPV. For instance, in a study of factors buffering the association between hazardous alcohol use and IPV, higher pay grade was found to be protective [43]. Also, because career satisfaction is likely associated with career success, we considered whether it might be protective among other socioeconomic factors in this study. This variable was not considered by Elbogen et al. [40]. However, in developing a screening tool for family violence risk, Stith et al. [44] found that military satisfaction was a significant predictor. In another study, workplace satisfaction in the Air Force was predictive of reductions in both IPV perpetration and victimization [45]. Given the salience of socioeconomic factors in the DoD, we expected this dimension to have both direct and moderating protective effects, thereby reducing the associations of both alcohol dependence and negative affect with IPV perpetration.

**Psychosocial factors.** Based on Elbogen et al.'s [40] findings, we hypothesized that a sense of control over one's life (e.g., self-determination or self-mastery), social support, and service member global social functioning would be protective against IPV perpetration. Additionally, Thomsen et al. [46] suggested that posttraumatic growth (e.g., increased sense of meaning and value for life; appreciation of interpersonal relationships) may lower risk for family violence perpetration following combat exposure for some service members. Relationship-focused growth in particular may be important because traumatic experiences have the potential to negatively or positively impact relationship trust and the value placed on interpersonal connection [32,47]. Maintaining a strong sense of the importance of interpersonal connection or relationship values may reduce the likelihood that PTSD symptoms will lead to negative relationship outcomes such as IPV.

**Physical health-related factors.** Like Elbogen et al. [40], we evaluated the importance of healthy sleep and low levels of self-reported pain as protective factors against IPV. We expected higher total hours of sleep and lower pain levels would be protective [14]. Elbogen [40] also examined self-care as a third marker in this category, whereas we considered health-related quality of life (i.e., physical functioning), hypothesizing that this indicator would globally operationalize health-related adjustment.

## Current study

Although Elbogen et al. [40] identified three separate types of factors that may be protective against general aggression among veterans, they did not evaluate the unique contributions of each in multivariable analyses. Rather, they examined the cumulative influence of the total number of protective factors participants reported. Elbogen et al. [40] did test for moderation, which many resilience theorists argue is a defining characteristic of protective processes [48]. Specifically, protective processes should act as buffers, reducing the impact of risk factors when they are at high levels, while having minimal effects when risk factors are at low levels. Elbogen et al. found this pattern of moderated protective effects [40]. However, later theorists have expanded the concept of protective processes beyond buffering the impact of risk factors. Importantly, Masten [49] suggested protective factors also may have direct or mediated protective effects that reduce the likelihood of poor outcomes. Direct, mediated, and moderated protective effects all may interrupt the risk process through different mechanisms, offering unique opportunities for intervention. Therefore, in the current study our objective was to

extend previous findings by testing the unique direct, mediated, and moderated protective effects of socioeconomic, psychosocial, and health-related factors on IPV in a multivariate model including substance use and five-factor PTSS clusters as risk factors (see Fig 1).

We explored the hypothesis that hyperarousal would prospectively predict incidents of IPV among military personnel as reported to the DoD FAP Central Registry (H1). However, we expected the effects of hyperarousal to be fully mediated through associations with symptoms of negative affect (i.e., anger/irritability, trouble concentrating, and sleep problems). We further hypothesized that hyperarousal would have a larger or potentially exclusive effect on negative affect

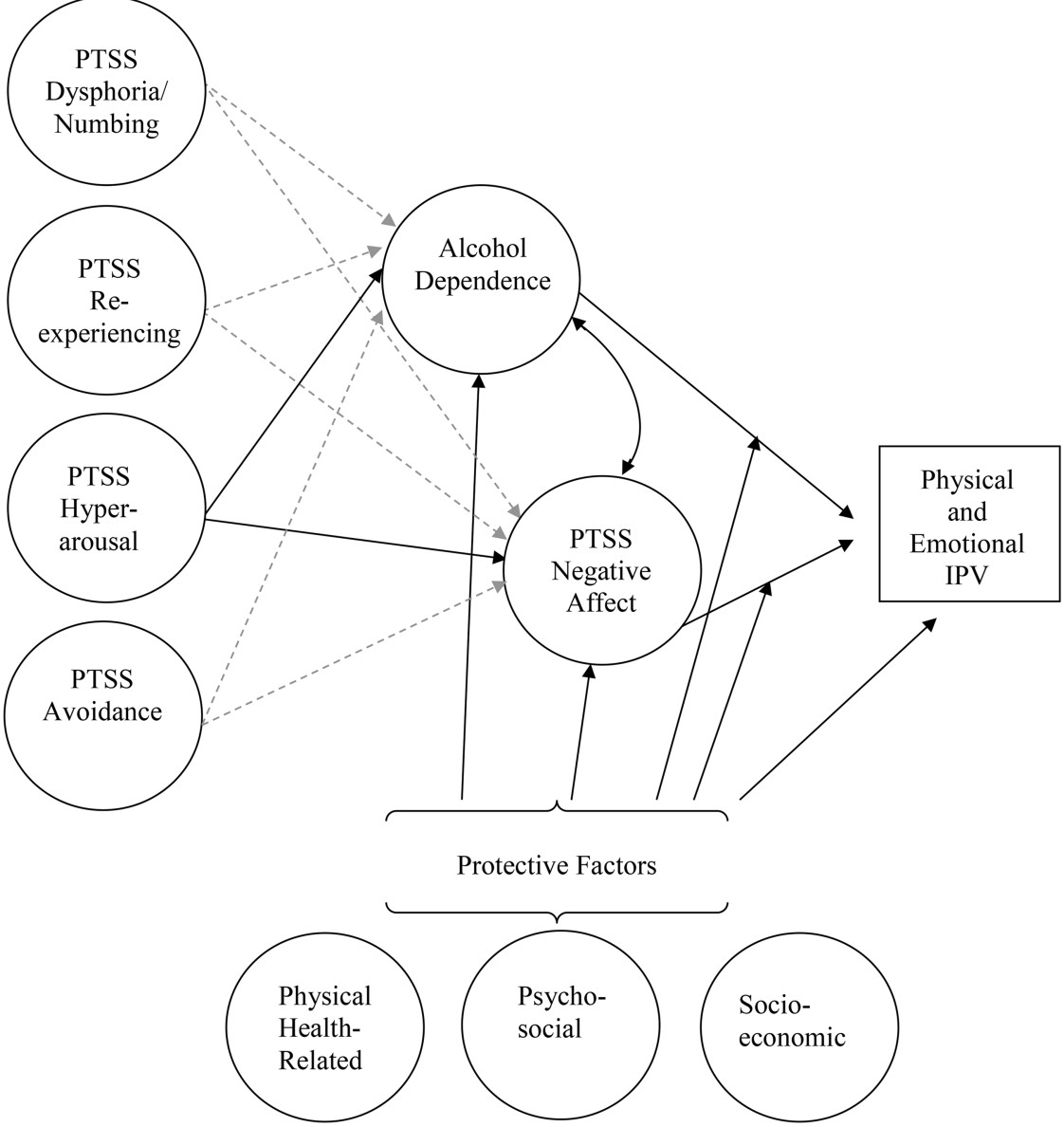

**Fig 1. Hypothesized structural equation model.** Model paths represented by dashed lines were hypothesized to be non-significant. PTSS = posttraumatic stress symptoms; IPV = intimate partner violence.

symptomology compared with other clusters of PTSS, such as dysphoria, re-experiencing, and avoidance (H2). We expected three types of protective factors (socioeconomic, psychosocial, and physical health) to predict decrements in IPV risk through multiple paths: direct effects (H3), indirect effects (mediation) through negative affect and alcohol dependence (H4), and moderating effects on the impact of alcohol dependence and negative affect on IPV risk (H5) (see Fig 1).

## Methods

### Participants

At the outset of the current study, data were available for 201,619 Millennium Cohort Study participants enrolled from all U.S. service branches and components in four consecutive panels from 2001 to 2013. After enrollment, participants are re-surveyed approximately every 3 years both during and after service life. For additional methodological details about this population-based study, see Ryan et al. [50]. Participants from all recruitment panels who reported they were on active duty (versus separated or Reserve/Guard) when responding to the Millennium Cohort Study's 4th data collection cycle, open from April 26, 2011 to April 4, 2013, were eligible (N = 54,667), and all survey data analyzed were from this survey cycle. DoD FAP Central Registry data on family violence perpetration are only available for those in an active-duty status, and the 2011–2013 survey cycle was defined as the baseline for this study to ensure the IPV incidents extracted for the outcome (May 23, 2011 to January 4, 2017) would be after the 2010 implementation of the current DoD criteria for confirming IPV incidents across services [51,52].

In the final sample, 74% were enlisted (26% officers), the mean age was 30.76 years (standard deviation [SD] = 7.08), and 27% were female. The majority (70%) of those in the final sample reported they were White non-Hispanic, with 11% Black non-Hispanic, 8% Hispanic, and 11% other. Most (53%) had a high school diploma or equivalent, while 47% had an associate, bachelor's, or graduate degree. There were 21% single (never married), 67% married, and 12% separated, divorced, or widowed participants. Also, 38% were Army, 35% Air Force, 18% Navy, and 9% Marine Corps personnel.

The Millennium Cohort Study was approved by the Naval Health Research Center Institutional Review Board in compliance with all applicable federal regulations governing the protection of human subjects (protocol number NHRC.2000.0007). Written or electronic informed consent was obtained from all participants. Coauthors affiliated with the Naval Health Research Center and listed on the approved human subjects protection protocol for the Millennium Cohort Study had access to subject identifiers for the purpose of archival data management and integration. Data were de-identified prior to analysis.

### Measures

**Intimate partner violence.** Each U.S. service branch is required to submit reports of domestic abuse perpetration and victimization (emotional, physical, sexual, neglect), as well as child abuse and neglect incidents, among active-duty personnel and their families to the DoD FAP Central Registry [13]. DoD FAP uses the term domestic abuse to encompass IPV incidents occurring in intimate relationships inclusive of those who are married, formerly married, current/former romatic partners who share(d) a domicile, or coparents who share biological offspring [53]. Within these types of intimate relationships, the DoD definition of domestic abuse comprises:

> Domestic violence [use, attempted use, or threatened use of force or violence against a person, or a violation of a lawful order issued for the protection of a person], or a pattern of behavior resulting in emotional or psychological abuse, economic control, or interference with personal liberty...[53]

Note that in comparison to the CDC definition of IPV [1,2], the FAP Central Registry domestic abuse data do not distinguish stalking or track incidents in dating relationships not involving cohabitation or a history of marriage or co-parenthood. Also, unlike some service-specific registries, the DoD FAP Central Registry does not retain individually

identifiable data for alleged incidents (reports that did not meet full definitional criteria for abuse). Incident reports are submitted to the Central Registry for intervention purposes independent of whether legal action is initiated. Central Registry records include the report date, type of incident (emotional, physical, or sexual abuse, as well as neglect), abuse severity ratings, and relationship between perpetrator and victim.

For the current study, a dichotomous variable was created to identify all participants with a report that met the current DoD criteria for an emotional or physical domestic abuse perpetration incident (i.e., "met criteria incidents") occurring any time after they completed their baseline survey. Sexual abuse and neglect reports were excluded due to low incidence rates. The observation period extended from survey completion date (earliest possible: April 26, 2011) to receipt of FAP data extract (September 1, 2017) or service separation, whichever came first. However, the first IPV outcome incident observed was reported May 23, 2011 and the last was January 4, 2017; it is possible that documentation for some incidents reported later was delayed and had not yet been recorded at the time of data extraction. Finally, to control for prior perpetration history, we identified participants with any incidents of IPV perpetration documented in the Central Registry prior to their 2011–2013 survey completion (observed date range: November 7, 1995 to October 16, 2012).

**Risk factors.** The 17-item PTSD Checklist (PCL) [54] assessed whether participants had experienced PTSS in the past month (1 = *not at all* to 5 = *extremely*). We created summed scores for symptom clusters representing a five-factor PTSD structure [38]. These included re-experiencing (five items; $M$ = 6.74, $SD$ = 3.50, Cronbach's alpha [$a$] =.93), avoidance (two items; $M$ = 2.62, $SD$ = 1.52, Pearson correlation [$r$] =.84), numbing (five items; $M$ = 6.96, $SD$ = 3.55, $a$ = .88), hyperarousal (two items; $M$ = 2.77, $SD$ = 1.61, $r$ = .73), and negative affect (three items; $M$ = 5.21, $SD$ = 2.79, $a$ = .82). Total symptoms reported (yes/no) on the four-item CAGE questionnaire [55] (i.e., feeling a need to cut down on drinking, feeling annoyed by comments about one's drinking, feeling guilty about drinking, or needing an "early morning drink") operationalized lifetime alcohol dependence ($M$ = 0.25, $SD$ = 0.67, $a$ = .68).

**Socioeconomic protective factors.** We considered five different socioeconomic indices for this study. On the 2011 Millennium Cohort survey, participants were asked to (a) report their education level on a 6-point scale (less than high school, high school/general equivalency diploma, some college, associate degree, bachelor's degree, graduate/professional degree; $M$ = 3.81, $SD$ = 1.29), (b) rate their current career satisfaction on a 5-point scale ($M$ = 3.91, $SD$ = 1.17), and (c) rate the level of their "financial problems or worries" on a 3-point scale (reverse scored; 0 = *bothered a lot* to 2 = *not bothered; M* = 1.57, $SD$ = 0.63) in response to a single item from the Patient Health Questionnaire (PHQ) [56]. Two additional indices were derived from military personnel records available through the Defense Manpower Data Center (DMDC): military rank (officer vs. enlisted) and monthly basic pay ($M$ = 3504.92, $SD$ = 1661.78) estimated based on pay grade and length of service.

**Psychosocial protective factors.** Psychosocial indices included the two-item Social Functioning scale from the Veterans RAND 36-Item Health Survey (VR-36), developed for the Medical Outcomes Study [56]. This scale assesses social limitations due to emotional or physical health problems. Scores were computed using manualized procedures for the VR-36 [57], with a sample $M$ = 85.61 and sample $SD$ = 22.26 ($r$ = .82). Higher values represent better social functioning. Self-mastery was assessed by summing responses (1 = *strongly disagree* to 5 = *strongly agree*) to three items from the Pearlin Self-Mastery Scale [58] assessing participants' sense of mastery and control over life and future goal attainment (*M* = 11.78, *SD* = 2.15, α = .63). Social support was assessed with a single item from the PHQ [59] that asks participants how much they are bothered by a lack of social support (i.e., having no one to turn to). This item was reverse scored (0 = *bothered a lot* to 2 = *not bothered*; $M$ = 1.76, $SD$ = 0.53) so that higher values would reflect greater social support. Lastly, a modified version of the 3-item relationship subscale (i.e., a sense of closeness with others, a sense of compassion, a belief that people are wonderful) from the Posttraumatic Growth Inventory (PTGI) [60] was included on the Millennium Cohort survey. These PTGI items were revised to assess overall strength and positivity (0 = *not at all* to 5 = *a very great degree*) rather than self-estimates of change in response to a traumatic event [61]. This subscale (*M* = 10.34, *SD* = 3.40, *a* = .83) was the final indicator representing psychosocial protective factors.

**Physical health-related protective factors.** Two additional scales from the VR-36 [56] were used as indices of health-related protective factors. These were the five-item general health scale ($M = 74.30$, $SD = 19.46$, $\alpha = .81$) and the two-item physical pain scale ($M = 73.62$, $SD = 22.88$, $r = .76$). Lastly, this category included responses to a single question about participants' average daily sleep: "Over the past month, how many hours of sleep did you get in an average 24-hour period?" ($M = 6.33$, $SD = 1.38$).

**Covariates.** Demographic and military characteristics were primarily derived from DMDC records (i.e., sex, age, race/ethnicity, and military service branch), but marital status (married, single, never married, divorced/widowed/separated) was self-reported at baseline. Analyses also controlled for any met criteria reports of IPV in the Central Registry that occurred prior to baseline. Last, due to variability in the time of baseline survey completion and because some participants left active service during the observation period (April 26, 2011 to September 1, 2017), we used DMDC service separation records to estimate total days of observation for each participant ($M = 1,591.26$, $SD = 724.78$, Range $= 2,427$) as an additional control variable.

## Analytic strategy

Initial analyses were conducted with IBM SPSS Amos [62]. We first estimated the overall rate of IPV perpetration (i.e., any DoD FAP met-criteria incident) by Millennium Cohort service members following their baseline 2011–2013 survey participation. Next, we computed unadjusted odds ratios representing the association of each baseline predictor and covariate with the likelihood of subsequent IPV perpetration. In initial analyses, we used case-wise deletion when we encountered missing data among predictors in our models (range $= 0$%–1.5% missing), and sample sizes were reported individually. However, to evaluate the hypothesized multivariate model shown in Fig 1, we used Mplus [63] with full information maximum likelihood estimation to manage missing data and applied the logit procedure due to the binary outcome. We further used Mplus to analyze a multi-group structural model testing structural invariance and sex differences in the associations between variables in Fig 1. Although not shown in Figs 1 or 2 (for simplicity), all mediational path analyses controlled for age, race/ethnicity, marital status, DoD service branch, any IPV reports prior to baseline, and days of observation post baseline. We also specified covariances between each interaction and the error terms for the mediating variables they moderated.

Common fit indices (e.g., Root Mean Squared Error of Approximation [RMSEA] or Comparative Fit Index [CFI]) are not optimal for logit modeling, because they assume continuous, multivariate normal data, and common alternatives do not offer vetted criteria for model rejection [64]. Therefore, we examined the Akaike information criterion (AIC) as a relative fit index to evaluate the adequacy of a series of nested models in representing the data; better models yield relatively lower AIC values. To select the best model, we further calculated delta AICs (ΔAICs), Akaike weights, and evidence ratios following guidance outlined by Burnham and Anderson [65] and Fabozzi et al. [66]. Specifically, ΔAIC compares AIC values for the model with the lowest AIC versus each alternative model; differences ≥2 support rejection of alternatives. Akaike weights represent the probability that a given model is the best fitting among a set of alternative models. Akaike weights can range from 0.00 to 1.00 and values > .50 support retaining (versus rejecting) a given model. Evidence ratios compare the extent to which the model with the lowest AIC fits better than an alternative model. The interpretation of evidence ratios is similar to odds ratios. For example, a value of 2 indicates that the minimum AIC model is two times more likely to be the best fitting relative to the alternative model. Evidence from all three criteria was used to select the final optimal model.

To develop the measurement model for Fig 1, all continuous variables were standardized and then latent functions were constructed separately for each factor using discriminant function analysis. This method is most appropriate because the indicators for both the PTSS clusters and the protective domains were conceptualized as composite-formative factors rather than as reflective indicators of a hypothesized latent construct [67,68]. Specifically, we expected each set/cluster to have a systemic synergy with a common impact on IPV. Our aim was to extract a factor representing the summary effect

of each set of intercorrelated variables, and to ensure that each composite factor represented the common variability from each set that was maximally associated with our outcome. We also were interested in descriptive results for factor loadings from these analyses, considering loadings above .32 to be substantive [69]. Discriminant scores were saved as observed variables and then used in structural analysis.

## Results

In FAP Central Registry records, we found at least one reported incident meeting DoD criteria for emotional or physical domestic abuse perpetration after baseline survey completion for 501 of 54,667 eligible participants. This represented approximately 9/1000 or just under 1% of the total sample, and it included 411 out of 40,106 men (1.0%) and 90 out of 14,561 women (0.6%). Table 1 lists unadjusted odds ratios for the bivariate association between the likelihood of a FAP incident and each covariate and hypothesized predictor. The table includes standardized odds ratios representing the change in odds of a FAP report associated with a one standard deviation increase in the value of each independent predictor. The table also includes maximum odds ratios representing the change in odds comparing participants with the lowest and highest observed values for each predictor. Bivariate associations were statistically significant for all predictors except self-mastery, which was dropped from further analyses.

### Measurement model

Because we conceptualized our predictors as formative composites rather than latent constructs, we used discriminant function analysis to develop the measurement model for our structural modeling. Table 2 lists the discriminant function loadings for the PCL symptom clusters, alcohol dependence (CAGE), and the three types of protective factors (socioeconomic, psychosocial, and physical health related). Initially, loadings for all but one of the variables were above .32. However, within the socioeconomic factor, the loading for military satisfaction was only .29. Therefore, career satisfaction was removed from this factor and entered in the model separately as an independent protective predictor. The variables with the largest loadings contributing to the final socioeconomic function were education level (.81) and basic pay (.79). Anchor variables with the largest loadings for the psychosocial and health-related protective factors were social support (.90) and average hours of sleep (.84), respectively. The largest loadings for the PTSS subscales were emotional reactivity to reminders of trauma (.93; re-experiencing), avoidance of mental reminders (.97; avoidance), feeling emotionally numb (.98; numbing), hypervigilance (.999) with a practical loading of unity on hyperarousal, and anger/irritability (.96; negative affect). Lastly, "feeling a need to cut down on drinking" had the largest loading (.92) on the discriminant function for alcohol dependence.

### Structural model

In an initial structural model, we tested each of the hypothesized paths shown in Fig 1. Paths that were not significant were fixed to zero to create a parsimonious model (AIC=2980253.02). The non-significant paths removed included the direct effects of both the health and socioeconomic status protective factors on alcohol dependence. Direct effects on IPV from physical health, psychosocial factors, and career satisfaction also were non-significant and were dropped. Almost all hypothesized protective moderation effects (i.e., interactions between protective factors and risk factors) were non-significant; however, the effect of negative affect on IPV was moderated by socioeconomic status. Johnson-Neyman plots [70] were used to probe this interaction. Despite some variations in the strength of their effects, both negative affect and socioeconomic factors were significantly associated with IPV outcomes at all levels of moderation (i.e., 95% confidence intervals around the estimated effect of negative affect—and alternately socioeconomic status—on IPV risk plotted at all levels of the other variable did not contain zero). Contrary to expectations, the buffering effect of socioeconomic status was not amplified at higher levels of negative affect. Instead, the interaction pattern suggested floor effects for risk, and

**Table 1. Bivariate associations between predictors and the likelihood of any Family Advocacy Program (FAP) reports following baseline.**

| | Odds ratios | | | Odds ratios | |
|---|---|---|---|---|---|
| Variable (comparison) | Standardized[a] | Maximum[b] | Variable (comparison) | Standardized[a] | Maximum[b] |
| Covariates | | | Protective factors | | |
| Sex (male) | – | 0.60 p<.001 | Socioeconomic | | |
| Age, years | 0.64 p<.001 | 0.06 p<.001 | Education level | 0.55 p<.001 | 0.10 p<.001 |
| Race/ethnicity (White) | | | Military status (enlisted) | – | 0.19 p<.001 |
| Black, non-Hispanic | – | 2.90 p<.001 | Basic pay | 0.40 p<.001 | 0.00 p<.001 |
| Hispanic | – | 1.29 p=.127 | Career satisfaction | 0.85 p<.001 | 0.57 p<.001 |
| Other | – | 1.00 p=.991 | Financial stability | 0.72 p<.001 | 0.35 p<.001 |
| Marital status (married) | | | Psychosocial | | |
| Single, never married | – | 0.36 p<.001 | Self-mastery | 0.93 p=.140 | 0.68 p=.140 |
| Divorced/widowed/separated | – | 1.01 p=.940 | Relationship attitudes | 0.84 p<.001 | 0.45 p<.001 |
| Service branch (Army) | | | Social support | 0.75 p<.001 | 0.34 p<.001 |
| Navy | – | 0.72 p=.015 | Social functioning | 0.75 p<.001 | 0.28 p<.001 |
| Marine Corps | – | 0.80 p=.187 | Physical health related | | |
| Air Force | – | 0.82 p=.053 | Health quality of life | 0.84 p<.001 | 0.42 p<.001 |
| Any prior FAP incidents | – | 8.26 p<.001 | Pain level | 0.80 p<.001 | 0.38 p<.001 |
| Observation time, days | 0.94 p=.141 | 0.81 p=.141 | Average hours of sleep | 0.76 p<.001 | 0.04 p<.001 |
| Risk factors | | | | | |
| PTSS re-experiencing | 1.29 p<.001 | 4.32 p<.001 | | | |
| PTSS avoidance | 1.28 p<.001 | 3.72 p<.001 | | | |
| PTSS numbing | 1.29 p<.001 | 4.23 p<.001 | | | |
| PTSS hyperarousal | 1.29 p<.001 | 3.57 p<.001 | | | |
| PTSS negative affect | 1.35 p<.001 | 3.69 p<.001 | | | |
| Alcohol dependence | 1.17 p<.001 | 2.62 p<.001 | | | |

*Note.* N = 53,915–54,667. PTSS = posttraumatic stress symptoms.

[a]Standardized odds ratios estimate the change in likelihood of a report given a one standard deviation difference in scores for each variable (not applicable for categorical predictors). [b]Maximum odds ratios estimate change in likelihood comparing participants with the lowest versus highest scores.

the observed pattern may be more interpretable if socioeconomic status is viewed as two ends of a spectrum that is protective at one end but presents risk at the other. Specifically, the risks posed by high negative affect and low socioeconomic status were diminished among those reporting more extreme levels of both (subadditive cumulative risk). Still, low socioeconomic status was associated with a fairly sizable increase in risk of IPV overall, with the largest direct effect in our model (see Fig 2).

We next tested the hypothesis that hyperarousal would be the only PTSS cluster to significantly predict both mediators (i.e., alcohol dependence and negative affect). Specifically, we evaluated whether model fit improved after adding any of the dashed paths in Fig 1 representing mediated effects from the other PTSS clusters. Each path was significant upon entry and AIC decreased substantially across models (ΔAICs ≥ 27.34). In a relaxed model freely estimating all dashed paths, all were significant except the path from re-experiencing to alcohol dependence. The relaxed model further decreased AIC (AIC = 2975583.91) and was the best fitting model relative to all former models (ΔAICs ≥ 1345.62, Akaike weight = 1.00, evidence ratios > 1000.00). Because the path from re-experiencing to alcohol dependence was not significant, an additional model was estimated constraining that path to zero (AIC = 2975584.54). This model was statistically equivalent to the relaxed model (ΔAIC = 0.63, Akaike weight = 0.42, evidence ratio = 1.37). However, we

**Table 2. Discriminant function loadings for the final measurement model.**

| Indicator | Loading | Indicator | Loading |
|---|---|---|---|
| Risk factors | | Protective factors | |
| Re-experiencing (PCL) | | Socioeconomic | |
| Emotionally reactive | .93 | Education level | .81 |
| Recurrent dreams | .84 | Basic pay | .79 |
| Physically reactive | .84 | Military status (enlisted) | .72 |
| Flashbacks | .76 | Financial stability | .63 |
| Intrusive thoughts | .74 | Psychosocial | |
| Avoidance (PCL) | | Social support (PHQ) | .90 |
| Mental reminders | .97 | Social functioning (VR-36) | .85 |
| Situational reminders | .95 | Relationship strengths (mPTGI) | .45 |
| Numbing (PCL) | | Physical health | |
| Emotionally numb | .98 | Sleep (average hours) | .84 |
| Distant from others | .78 | Bodily pain (VR-36) | .75 |
| Loss of interest | .75 | General health (VR-36) | .59 |
| Poor memory of trauma | .64 | | |
| Foreshortened future | .64 | | |
| Hyperarousal (PCL) | | | |
| Hypervigilance | .99 | | |
| Exaggerated startle | .77 | | |
| Negative affect (PCL) | | | |
| Anger/irritability | .96 | | |
| Sleep disturbance | .75 | | |
| Difficulty concentrating | .60 | | |
| Alcohol dependence (CAGE) | | | |
| Cut down | .92 | | |
| Guilty feelings | .68 | | |
| Annoyed by criticism | .62 | | |
| Eye opener | .51 | | |

*Note*. PCL = PTSD Checklist; PHQ = Patient Health Questionnaire; VR-36 = Veterans RAND 36-Item Health Survey; mPTGI = Modified Posttraumatic Growth Inventory.

made the interim decision to retain the relaxed model to test sex differences in the effect of re-experiencing on alcohol dependence.

Next, we specified a relaxed multi-group model to evaluate sex invariance across structural associations, allowing all paths to freely vary across groups (AIC = 3030967.30). We compared that model to one with all paths constrained to be equal across groups (AIC = 3031148.56). The free model fit significantly better than the constrained model (ΔAIC = 181.27, Akaike weight = 1.00, evidence ratio > 1000.00), suggesting moderation by sex. We then estimated a series of models to verify specific sex differences. Each model serially specified a group equality constraint for one of the hypothesized paths; poorer fitting models indicated that the effect size for a specific constrained path differed between groups.

Several models indicated worse fit when equality constraints across sexes were imposed (see Fig 2 for final effect sizes). Associations between the four exogenous PTSS clusters and alcohol dependence were invariant by sex except for PTSS dysphoria/numbing; this path was stronger for males than for females based on two out of three criteria. The

ΔAIC was < 2 (1.19), however the small Akaike weight (0.24) and considerable evidence ratio (1.81) suggested worse fit for the constrained model. In contrast, the associations between all but one of the exogenous PTSS clusters and negative affect varied by sex. Two were stronger for male participants, including the paths to negative affect from dysphoria/numbing (ΔAIC = 12.11, Akaike weight = 0.00, evidence ratio = 426.03) and hyperarousal (ΔAIC = 31.58, Akaike weight = 0.00, evidence ratio > 1000.00). In contrast, the path from avoidance to negative affect was stronger for female participants (ΔAIC = 6.32, Akaike weight = 0.02, evidence ratio = 23.56); however, this effect was quite small overall and was statistically significant only for females.

Only two protective factors were significantly associated with reduced alcohol dependence: career satisfaction and psychosocial factors. The strength of this protective association did not vary by sex for career satisfaction, whereas psychosocial factors predicted greater reductions in alcohol dependence for females than for males (ΔAIC = 37.41, Akaike weight = 0.00, evidence ratio > 1000.00). All protective factors were significantly predictive of lower negative affect. For both career satisfaction and socioeconomic factors, the strength of the association was similar regardless of sex. However, the effect of health-related factors on negative affect was stronger for males (ΔAIC = 56.09, Akaike weight = 0.00, evidence ratio > 1000.00), while the protective influence of psychosocial factors was stronger for females (ΔAIC = 84.00, Akaike weight = 0.00, evidence ratio > 1000.00).

Notably, all direct effects on IPV perpetration in our model were invariant across sex. Further, across mediated paths, only the magnitude of the effects differed for males and females; all structural associations were in the same direction for both sexes, and those significant for one group were generally significant for the other. The one exception was the path between avoidance and negative affect, which was significant only for females.

A final multi-group path analysis was estimated to constrain statistically equivalent paths across groups and allow only those that significantly differed to vary (AIC = 3030957.70). This model was compared with the fully relaxed multi-group model (AIC = 3030967.30). The final model fit better than the fully relaxed multi-group model (ΔAIC = 9.60, Akaike weight = 0.99, evidence ratio = 121.28) and was therefore retained. The final multi-group model explained a significant ($ps < .001$) proportion of the variance ($R^2$) in alcohol dependence (male = .08; female = .07), negative affect (male = .57; female = .55), and IPV (male = .25; female = .27). For final standardized regression coefficients and $p$-values, see Fig 2. For summary effect sizes and the significance of total effects (combining direct and indirect paths for each predictor) and specific indirect effects in the final model, see S1 and S2 Tables, respectively. All distal predictors with significant paths to negative affect had significant mediated associations ($p < .05$) with IPV risk. However, for two of the distal predictors with significant paths to alcohol dependence (i.e., hyperarousal and career satisfaction), the mediated path to IPV risk did not reach significance.

### Sensitivity analysis

Sleep was potentially confounded in our models because sleep difficulties were a component of the PTSS negative affect cluster, while more hours of sleep also was hypothesized as a component of health-related protective factors. Therefore, we conducted a sensitivity analysis to determine whether this inflated the observed association between the two constructs. Specifically, we respecified the final multi-group model in three separate ways: (1) removed average hours of sleep from the health-related protective factors construct, (2) removed sleep disturbances from the PTSS negative affect cluster, and (3) replaced the PTSS negative affect cluster with the PCL's anger/irritability item. In all three respecified models, effect sizes for the associations between adjusted health-related and PTSS negative affect factors were smaller than in the original model (Alternative 1 [A1]: male = −.16, female = −.13; A2: male = −.12, female = −.07; A3: male = −.12, female = −.07), but the significant associations among variables were consistent with the final multi-group model presented in Fig 2. Therefore, although the inclusion of sleep health in both constructs influenced effect sizes, it did not impact the structure of the model.

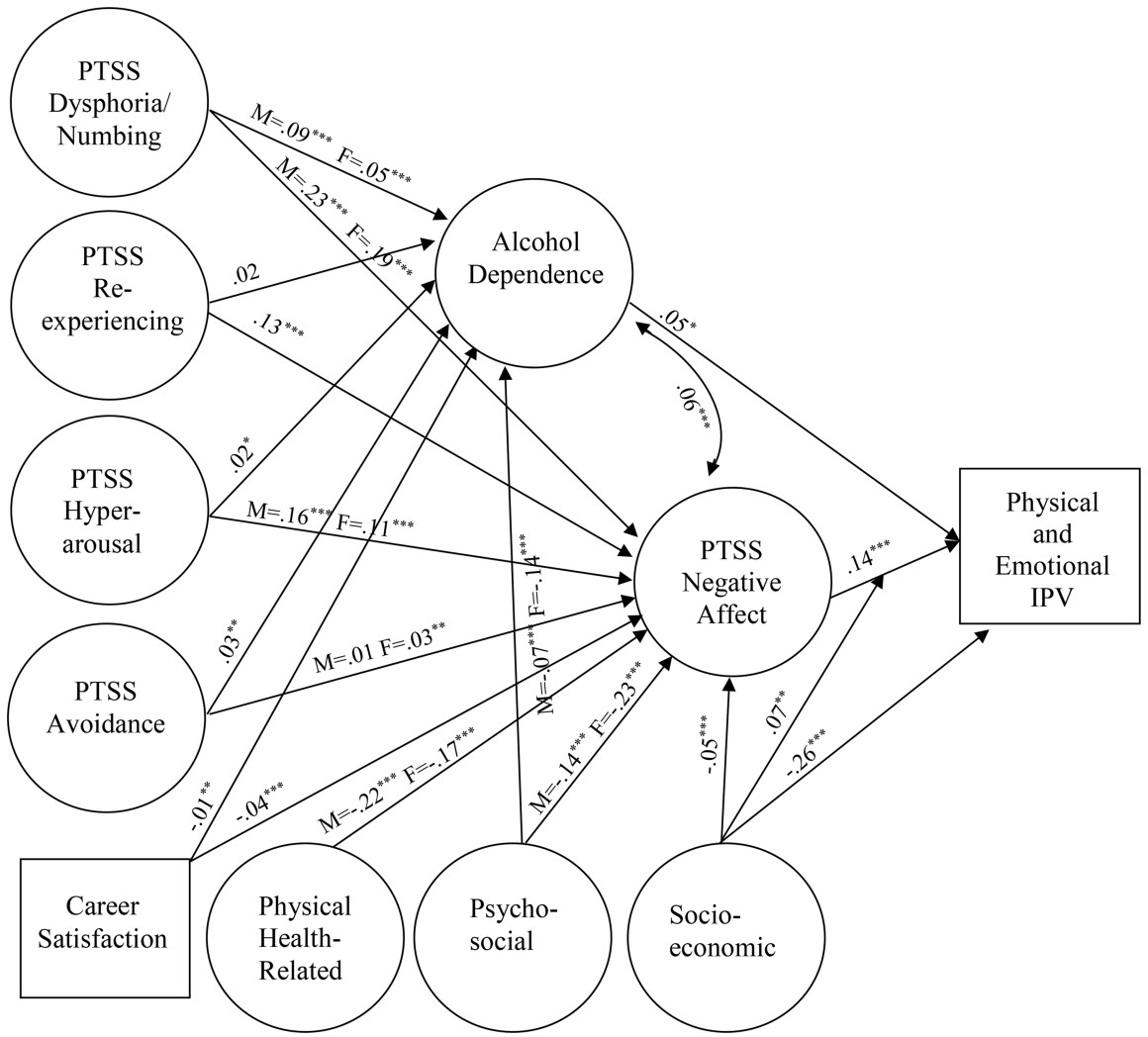

**Fig 2. Final structural equation model.** Paths labeled with one coefficient were statistically equivalent between groups, whereas paths with two coefficients (one for each group) were moderated by sex. M = male group ($N$ = 40,106); F = female group ($N$ = 14,561); PTSS = posttraumatic stress symptoms; IPV = intimate partner violence. *$p$ < .05. **$p$ < .01. ***$p$ < .001.

## Discussion

This study explored predictors of reported incidents of IPV among active-duty personnel meeting criteria for domestic abuse in the DoD FAP Central Registry in order to develop a model of risk and protective factors for IPV perpetration. Using longitudinal data from participants in the Millennium Cohort Study, we evaluated alcohol dependence and PTSS negative affect as mediators of the impact of other trauma-related symptoms on IPV perpetration. We also evaluated the extent to which these effects may be buffered by three categories of protective factors: socioeconomic, psychosocial, and physical health. We found significant support for three of five initial hypotheses. Our first hypothesis was fully supported; hyperarousal symptoms predicted increased risk of IPV perpetration. Also, this association was fully mediated by PTSS-related negative affect (H2). However, hyperarousal was not more strongly associated with negative affect than the other PTSD symptom clusters. The hypotheses that our three types of protective factors would have both direct (H3) and

mediated (H4) effects on IPV risk were partially supported (see Fig 2). However, none had buffering moderation effects (H5) on the direct associations between negative affect or alcohol dependence and IPV. Although the interaction between socioeconomic factors and negative affect was significant, the pattern we observed was more indicative of floor effects for risk than of a protective process.

In contrast to prior research and theory [15,28], among the PTSD symptom clusters, hyperarousal PTSS were not uniquely predictive of IPV (H2). Hyperarousal had a significant effect on IPV mediated through negative affect, but numbing, re-experiencing, and avoidance did as well. Notably, the size of the mediated effects for both numbing and re-experiencing were as large or larger than those for hyperarousal. Interestingly, the significant effect of hyperarousal on negative affect was moderated by sex, with a smaller potentiating effect for females than males. A similar pattern was noted for the dysphoria/ numbing cluster, suggesting both of these clusters may heighten risk somewhat more for males than females.

In general, the mediated effects of the PTSS clusters on IPV were significant for both males and females. However, the avoidance cluster was the weakest predictor of negative affect overall and was significant only for females. Also, the direct effects of alcohol dependence and negative affect on IPV were not moderated by sex. This suggests the final structure of our model was generally applicable to males and females, and that multiple PTSS clusters exacerbated negative affect for both. Importantly, in our final model, we did not replicate a suppressor effect (i.e., protective trend for numbing in fully adjusted analyses among women) noted in our prior work exploring sex differences in the associations of PTSS clusters with IPV perpetration risk [14]. However, our prior models did not account for the mediating role of either alcohol dependence or negative affect; this highlights the importance of properly defining the associations among these predictors to accurately specify a predictive model of IPV perpetration applicable to both male and female service members.

The cumulative impact of the predictors in our model on IPV perpetration risk accounted for just 25%–27% of the variance in FAP incidents for males and females, respectively. It is likely that multiple additional factors must be considered to fully predict this outcome. Furthermore, the direct effects of both alcohol dependence and PTSS negative affect on IPV risk were smaller here than has been observed in other studies [15]. Several factors may have contributed to this [14]. Importantly, effect sizes related to PTSD and other mental health disorders often have been smaller in community samples than in clinical samples [15]. Also, operationalizing IPV incidents using DoD FAP Central Registry data may have impacted effect sizes. Only 1% of our participants had documented IPV incidents during our observation period. Both underreporting and inaccuracies in Central Registry data [71] may create noise, diminishing effect sizes. There also may be differences in risk and protective factors for self-reported versus Central Registry-documented incidents of IPV. Future research should compare the performance of predictive models of IPV perpetration when measuring outcomes via self-reports versus official reports of IPV incidents.

Interestingly, socioeconomic status played a prominent protective role in our final model and had the only direct protective effect on risk of a FAP incident. This central role of socioeconomic factors in predicting IPV perpetration risk is consistent with prior research [72]. In both civilian and military contexts, financial well-being and social status provide protective resources that can reduce risk for multiple poor outcomes, including IPV perpetration. However, the moderating effect we noted with respect to this dimension was not buffering (i.e., producing a larger protective effect at higher reported levels of negative affect) as Elbogen et al. [40] found. This may not be too surprising; in a seminal review, Masten [49] noted that moderating protective processes are rare and inconsistently observed across studies. Furthermore, Masten suggested that some factors such as socioeconomic status are complex and may offer a protective effect at one end of the continuum but increase risk at the other, with potentially different mechanisms facilitating these effects at opposing poles. The interaction pattern we found was more indicative of floor effects for risk, with the combined effects of low socioeconomic status and greater negative affect having less impact on IPV perpetration than expected based on their individual effects.

The protective effects of the psychosocial resources were fully mediated and were more pronounced for females than males with respect to both alcohol dependence and negative affect. This suggests a greater indirect protective effect for females, although the relationship was significant for both sexes. Social support had the strongest loading among our

psychosocial indices, and the DoD offers programs to bolster social support for military families, particularly during deployment or permanent changes of duty station [73,74]. This is important for this highly mobile population, as personnel and their families must frequently reestablish support networks. Further, tools for maintaining long-distance connections with family and friends through social media continue to proliferate [75], which may help military families in the longer term. Going forward, it will be important to study how the means military families use to maintain social support networks continue evolving, and if the mode of social support may make a difference in its effects as a protective factor against IPV.

Sleep had the largest loading on the discriminant function for health-related factors, and health-related factors had an indirect protective effect on IPV perpetration risk, mediated through negative affect. In this case, the effect was stronger for males than females. Surprisingly, few studies of military or civilian samples have evaluated the impact of sleep on IPV perpetration [76–78], and it is likely another complex factor, with healthy sleep reducing risk while sleep problems increase risk. As a risk factor, the synergy of poor sleep and anger/irritability has been a predictor of aggression in prior research [78]. Among those experiencing PTSD, this may be uniquely important, given that nightmares and sleep disruption are common symptoms. Conversely, as a protective factor, healthy sleep is essential for the cognitive processing necessary to inhibit reflexive aggression. In one recent study of a community sample, relationship aggression was associated with multiple aspects of sleep quality mediated by self-control [76]. Sleep problems also likely exacerbate physical (e.g., chronic pain) and mental health symptoms and may lead to interrelated mental and behavioral conditions [78,79]. These associations make sleep a potentially powerful common predictive factor, particularly since sleep problems and sleep medication use are commonly reported by military personnel and have been associated with deployment [80]. Studies of how sleep education may interrupt different mechanisms of risk and serve as a prevention tool to reduce a range of negative outcomes, including IPV perpetration, are warranted.

One factor that was poorly conceptualized in our original hypothesized model was career satisfaction. Although we originally assumed that career satisfaction would cluster with other indicators of socioeconomic protective factors, it did not. Career satisfaction was therefore entered separately as an independent protective predictor in our final model. Interestingly, career satisfaction had small but significant protective effects on both mediators, although its only impact on IPV risk was mediated through negative affect (i.e., the mediated path via alcohol dependence was not significant). These results are consistent with prior research suggesting career satisfaction may be useful in screening for IPV risk [44,45]. Career satisfaction may be an orphan indicator for an additional protective dimension poorly defined in our model. Future research should further explore career satisfaction— along with other likely correlated military factors, such as unit cohesion, leadership quality, leadership support for families, and work–family balance [72]—as additional potential protective factors.

### Strengths and limitations

This study had a number of strengths and limitations. Use of FAP reports meeting criteria for physical and emotional domestic abuse to operationalize IPV perpetration provided an objective assessment. However, use of these data restricted generalizability, because FAP data do not include incidents in dating relationships or identify stalking incidents. Additionally, official reports of domestic abuse underrepresent the full scope of the problem and are likely to disproportionately represent more severe incidents. The percentage of our sample with met criteria incidents was comparable to incidence rates reported by the FAP [81], but the incidence is low and likely limited our power to detect very small effects. The large sample of over 50,000 participants helped to offset this statistical limitation. Nonetheless, it will be helpful to compare the current results with findings based on populations with higher IPV prevalences or more sensitive operationalizations (e.g., self-reports). This study solely focused on developing a model for perpetration risk. It would be helpful in the future, though, to further examine both IPV victimization and perpetration and compare risk and protective factors for bidirectional versus unidirectional IPV. Although beyond the scope of this paper, the Millennium Cohort Study recently added a screening measure for IPV [82], making these types of analyses possible in future research [83].

The Millennium Cohort survey includes many standardized assessments for risk and protective factors that we were able to use in our analyses, but there were some limitations in the available measures as well. First, the items assessing self-mastery were only a subset of the Pearlin Self-Mastery Scale [58]. This may have contributed to the lack of association between this construct and IPV perpetration in our analyses. Another limitation is that our operationalization of military basic pay for active-duty participants was estimated from pay grade and time in service. Information about how the U.S. military sets pay rates based on these factors is publicly available. However, our estimates could not take into consideration pay bonuses or augmented special pay participants may have received. Also, the assessment of PTSS in this study was based on the 17-symptom criteria for PTSD included in the Diagnostic and Statistical Manual of Mental Disorders, 4th Edition (DSM-IV) rather than on the current DSM-5 criteria. Despite these limitations, the Millennium Cohort Study offers a unique, large, population-based military sample of active-duty personnel from all service branches, so our findings can be generalized across the active-duty DoD community.

## Implications

Our results can inform future research on IPV in several ways, but one important implication is the advisability of using a five-factor structure for PTSD symptomology in studies of IPV risk. Elhai's [38] five-factor structure eliminates a confound between hyperarousal and negative affect, and distinguishing them in this study allowed us to better explore and confirm the role of negative affect as a mediator between PTSS and IPV perpetration risk. Also, the use of a five-factor structure is likely important in establishing models that are applicable to both males and females, as the effects of multiple clusters, including hyperarousal, were moderated by sex.

In dealing with a range of challenging issues such as IPV, sexual assault, and suicide, the DoD has been increasingly emphasizing integrated and holistic approaches to prevention and response, through understanding and leveraging modifiable protective factors [84]. This trend has followed in the wake of an increasing awareness of the interconnectedness of many poor behavioral health outcomes and greater focus on common factors approaches to prevention and intervention [81,85]. Efforts like the current study that explore modifiable protective factors are helpful in informing programs from this perspective. Our findings suggest future program development and evaluation should consider whether common protective factors such as socioeconomics (e.g., education level and economic stability), health factors (e.g., sleep quantity and quality), and psychosocial factors (e.g., social support) can be modified at a population level to reduce risk for multiple interrelated negative outcomes (e.g., PTSD, substance dependence, and IPV).

Given the mobility of the military population, helping personnel maintain both formal and informal support networks is an important challenge, and one which the DoD has worked to address for many years [42]. Recent policy shifts have aimed to reduce military mobility [86]; if implemented extensively, this could help stabilize social networks and support resources for those in service. Furthermore, social integration (e.g., unit cohesion) within the DoD is not only important for readiness [87], but also for integrated prevention [88]. This may have implications, particularly for women, among whom our results suggest social support may be even more critical. Unfortunately, evidence also suggests that servicewomen may struggle more than servicemen with both integration in the DoD community [89] and maintenance of interpersonal relationships outside the work environment [90].

The prevalence of incidents reported to the DoD Central Registry meeting criteria for domestic abuse among participants in this study was low compared to self-report estimates. For instance, a recent review of self-reported IPV among service members and veterans indicated that more than 1 in 10 may have perpetrated as recently as the past year, while 2 in 10 may have experienced IPV [91]. Furthermore, research suggests that the majority of IPV is bidirectional. In military populations, as many as three quarters of the couples experiencing IPV may be engaged in bi-directional aggression [83]. This means most often personnel perpetrating IPV are also experiencing it and living in an environment that may exacerbate trauma symptoms [92]. Approaches to treatment in the DoD, therefore, need to be sensitive to the systemic dynamic of relationships and maintain a trauma focus. Encouragingly, there are programs like this, such as the trauma-focused

intervention Strength at Home developed within the VA; a recent trial of the couples version of this program among military personnel found reductions in IPV perpetration among both service members and their spouses [93]. Further implementation of programs like this in the DoD could help both reduce IPV perpetration and improve continuity of care during and after service.

## Supporting information

**S1 Table. Total effects of all variables on intimate partner violence in the final structural equation model.** *Note*. SE = standard error; FAP = Family Advocacy Program; PTSS = posttraumatic stress symptoms. Male group $N = 40,106$; Female group $N = 14,561$. *$p < .05$. **$p < .01$. ***$p < .001$.
(DOCX)

**S2 Table. Indirect effects for the final structural equation model.** *Note*. Coeff = unstandardized coefficient; LLCI = 95% lower limit for confidence interval; ULCI = 95% upper limit for confidence interval; AD = alcohol dependence; IPV = intimate partner violence; NA = negative affect; CS = career satisfaction; HR = health-related protective factors. Confidence intervals are based on 1,000 bootstrapped samples. Bolded confidence intervals are non-significant (i.e., encompass zero). Male group $N = 40,106$; Female group $N = 14,561$. *$p < .05$. **$p < .01$. ***$p < .001$.
(DOCX)

## Acknowledgments

The authors express profound gratitude to the study participants who contribute their time to the Millennium Cohort Program. We also sincerely thank the DoD Family Advocacy Program for providing access to Central Registry data and supporting the study team in using these data.

Disclaimer: VAS is an employee of the U.S. Government. This work was prepared as part of her official duties. Title 17, U.S.C. §105 provides that copyright protection under this title is not available for any work of the U.S. Government. Title 17, U.S.C. §101 defines a U.S. Government work as work prepared by a military service member or employee of the U.S. Government as part of that person's official duties. Report No. 25−23 was supported by the Defense Health Agency, Defense Health Program, Military Operational Medicine Research Program under work unit no. 60002. The funders had no role in study design, data collection and analysis, decision to publish, or preparation of the manuscript. The views expressed in this article are those of the authors and do not necessarily reflect the official policy or position of the Department of the Navy, Department of Defense, nor the U.S. Government. The study protocol was approved by the Naval Health Research Center Institutional Review Board in compliance with all applicable federal regulations governing the protection of human subjects. Research data were derived from approved Naval Health Research Center Institutional Review Board protocol number NHRC.2000.0007.

## Author contributions

**Conceptualization:** Valerie A. Stander, Sabrina M. Richardson, Kelly A. Woodall, Cynthia J. Thomsen, Joel S. Milner, James E. McCarroll, David S. Riggs, Stephen J. Cozza.

**Data curation:** Valerie A. Stander, Sabrina M. Richardson, Kelly A. Woodall.

**Formal analysis:** Valerie A. Stander, Travis N. Ray.

**Funding acquisition:** Valerie A. Stander.

**Methodology:** Valerie A. Stander, Sabrina M. Richardson, Kelly A. Woodall, Cynthia J. Thomsen, Joel S. Milner, James E. McCarroll, David S. Riggs, Stephen J. Cozza.

**Project administration:** Valerie A. Stander.

**Supervision:** Valerie A. Stander.

**Writing – original draft:** Valerie A. Stander, Travis N. Ray.

**Writing – review & editing:** Valerie A. Stander, Travis N. Ray, Sabrina M. Richardson, Kelly A. Woodall, Cynthia J. Thomsen, Joel S. Milner, James E. McCarroll, David S. Riggs, Stephen J. Cozza.

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
