## [Decision Letter · Decision Letter 0]

21 Oct 2025

Dear Dr. Stander,

Thank you for submitting your manuscript to PLOS ONE. After careful consideration, we feel that it has merit but does not fully meet PLOS ONE’s publication criteria as it currently stands. Therefore, we invite you to submit a revised version of the manuscript that addresses the points raised during the review process.

The topic is important, and the prospective design with SEM analyses has the potential to contribute to prevention/intervention science in military populations. However, several methodological and reporting elements are insufficiently justified or inadequately reported, which impedes interpretability, reproducibility, and the manuscript’s overall impact. Specifically, it is necessary to provide the rationale for key analytic decisions, report comprehensive SEM fit statistics, explicitly address the handling of a binary outcome within SEM, and offer a more nuanced theoretical framing. The Introduction and Discussion would benefit from deeper engagement with PTSD-IPV literature, clearer mediation considerations, and a more explicit discussion of limitations (notably, potential under-reporting in DoD IPVA data and DoD definition clarity). Minor editorial points (Table 1 p-values and more precise definitions) should be addressed to improve clarity and reproducibility.

We look forward to receiving your revised manuscript.

Kind regards,

Yordanis Enríquez Canto, Ph.D.

Academic Editor

PLOS ONE

Journal Requirements:

“Defense Health Agency, U.S. Department of Defense”

3. In the online submission form, you indicated that your data is available only on request from a third party. Please note that your Data Availability Statement is currently missing [the name of the third party contact or institution / contact details for the third party, such as an email address or a link to where data requests can be made]. Please update your statement with the missing information.

4. Please ensure that you include a title page within your main document. You should list all authors and all affiliations as per our author instructions and clearly indicate the corresponding author.

Reviewers' comments:

Reviewer's Responses to Questions

**Comments to the Author**

1. Is the manuscript technically sound, and do the data support the conclusions?

Reviewer #1: Yes

Reviewer #2: Yes

2. Has the statistical analysis been performed appropriately and rigorously?

Reviewer #1: Yes

Reviewer #2: Yes

3. Have the authors made all data underlying the findings in their manuscript fully available?

Reviewer #1: No

Reviewer #2: Yes

4. Is the manuscript presented in an intelligible fashion and written in standard English?

Reviewer #1: Yes

Reviewer #2: Yes

Reviewer #1: This study investigates modifiable risk and protective factors for intimate partner violence (IPV) perpetration among active duty U.S. military personnel, with the goal of informing prevention and intervention strategies. The authors employ a prospective path model using structural equation modeling (SEM), with IPV incidents—defined according to Department of Defense (DoD) criteria for psychological and physical abuse—as the primary outcome.

Key risk factors identified include posttraumatic stress symptoms (PTSS), alcohol dependence, younger age, enlisted status, and prior IPV incidents. Protective factors are categorized into three domains: socioeconomic, psychosocial, and physical health.

Minor Revisions Suggested:

1- Latent Variable Construction: Discriminant function analysis is not a conventional method for constructing latent variables in SEM. Confirmatory factor analysis (CFA) would be more standard and interpretable.

2- Model Fit Reporting: The manuscript should report a broader range of SEM fit indices (e.g., RMSEA, CFI, TLI), which are standard in SEM reporting and would enhance transparency.

3- Binary Outcome Modeling: While using a binary outcome (IPV) in SEM is valid, the manuscript lacks detail on whether robust estimators or logit link functions were used. This should be clarified.

4- Sensitivity Analysis: Effect size changes in the sensitivity analyses should be reported more explicitly to quantify the impact of model adjustments.

5- Methodological Justification: The rationale for selecting discriminant analysis over CFA should be clarified to help readers understand the methodological choices.

6- Table 1 Reporting: More precise p-values should be provided in Table 1 to improve interpretability and reproducibility.

Reviewer #2: This paper described research into the relationship between PTSD and IPV, highlighting potentially modifiable risk factors for perpetration. While the methodology and findings were largely clear, I have queries about the conceptualisation of the models in the main.

- While I agree with some of the hypotheses presented, it would be useful for further discussion of them in the introduction to better support the development and thinking behind the hypothesised model, for example, fuller presentation of research of PTSD clusters and potential impacts on IPV or relationship tension and of negative affect. This would help highlight why direct pathways between PTSS symptoms and IPVA were not included and the hypothesis of full mediation pursued given what is known of PTSD clusters and IPVA. I also query the inclusion of workplace satisfaction within socioeconomic factors. While satisfaction may be related to SES, one can have low SES and high job satisfaction. There is likely other research to support the point being made about links between SES with IPVA, especially given recruitment practices for the military. Stith and others may fit better as psychosocial factors.

- The discussion could include great reference to prior literature as well as an overview of the findings to better contextualise findings not only in terms of research and what is being added, supported or disputed, but what particular interventions could realistically be introduced? Some additional references needed in parts e.g., for the first line, page 6 line 116-118 regarding benefits of service etc.

- While not a major point, the large sample is extremely impressive but would not necessarily offset systematic under-reporting of IPV within DOD systems, especially when used as the outcome of interest (and a prevalence of only 1% is very low). This is unfortunately one of those limitations that must be embraced without current ability to address within the dataset until more robust measures of IPVA can be included. It would be useful to include DOD definitions of IPVA which are not explicitly stated making it hard to determine what exactly may have been included if it was in fact reported.

- Finally, there is also the point that, while not a focus here, those with mental health problems are perhaps at greater risk of experiencing IPVA than of perpetrating it.

**Do you want your identity to be public for this peer review?** For information about this choice, including consent withdrawal, please see our Privacy Policy

Reviewer #1: No

Reviewer #2: No

---

## [Author Response · Author response to Decision Letter 1]

9 Dec 2025

Thank you for the opportunity to revise and resubmit our article titled "Risk and protective factors for incidents of intimate partner violence among active duty military personnel" (PONE-D-25-34092). As requested, we are submitting a revised manuscript with both a clean copy and track changes version, as well as an itemized response to reviewers. As further requested, we have updated the file names for Figures 1 and 2; there are some minor edits in figure 2, so both clean and track change copies are submitted. However, there were no changes to Figure 1 other than to the file name. Finally, per instruction, we have updated the data availability statement in this online submission. Please let us know if there are any additional issues with this resubmission.

---

## [Decision Letter · Decision Letter 1]

17 Dec 2025

Dear Dr. Stander,

plosone@plos.org . A letter that responds to each point raised by the academic editor and reviewer(s). You should upload this letter as a separate file labeled 'Response to Reviewers'.A marked-up copy of your manuscript that highlights changes made to the original version. You should upload this as a separate file labeled 'Revised Manuscript with Track Changes'.An unmarked version of your revised paper without tracked changes. You should upload this as a separate file labeled 'Manuscript'.

We look forward to receiving your revised manuscript.

Kind regards,

Yordanis Enríquez Canto, Ph.D.

Academic Editor

PLOS One

Journal Requirements:

**Additional Editor Comments:**

Your revisions have substantially clarified the methods, with key decisions in the analytic approach now more clearly justified. Reviewer 2, who re‑evaluated the revised manuscript, is broadly positive about the changes and notes that only very minor issues remain before the paper is suitable for publication.

In light of the reviewer’s assessment and my own evaluation, I am pleased to inform you that I am prepared to recommend acceptance pending very minor revisions. Please address the following points in a brief final revision.

Reviewers' comments:

Reviewer's Responses to Questions

**Comments to the Author**

Reviewer #1: All comments have been addressed

Reviewer #2: (No Response)

2. Is the manuscript technically sound, and do the data support the conclusions?

Reviewer #1: (No Response)

Reviewer #2: Yes

3. Has the statistical analysis been performed appropriately and rigorously?

Reviewer #1: (No Response)

Reviewer #2: Yes

4. Have the authors made all data underlying the findings in their manuscript fully available?

Reviewer #1: (No Response)

Reviewer #2: Yes

5. Is the manuscript presented in an intelligible fashion and written in standard English?

Reviewer #1: (No Response)

Reviewer #2: Yes

Reviewer #1: The revisions are comprehensive and directly respond to the methodological and reporting concerns. The manuscript now provides clearer justification for analytic choices, improved transparency in reporting, and enhanced clarity in sensitivity analyses.

Reviewer #2: Thank you to the authors for their considered responses to reviewer comments. The methods in particular are much clearer and the decision made in analyses more robustly justified. There are some remaining comments that would further clarify and strengthen the paper but these are very minor.

1. There are some statements in the intro and overview of prior research that still need references (e.g., first few sentences, top of page 7 re anger and hyperarousal, sleep problems, end line 132). While SES is important, a nod to considering the role of military cultural factors in IPVA (eg work of Deirdre MacManus et al in UK) would also be useful even if not captured within the data.

2. I appreciate the inclusion of more discussion of issues of directionality and on the outcome measure itself. However, these should be earlier in the paper. The addition of the DOD definition is useful though it would be helpful for this to come earlier in the paper, or for a definition of IPVA to be added into the first paragraph and perhaps a note on any discrepancies between the two in the limitations. For example, it looks like the DOD definition doesn't include sexual IPVA (although this is mentioned later in the binary variable description) but wider IPVA definitions certainly would. The point on directionality could be made earlier also, methods perhaps, without undermining the paper as all data has limitations.

3. The gender sub-analyses were interesting but it would be useful to have an idea of how many women where in the dataset at this point.

**Do you want your identity to be public for this peer review?** For information about this choice, including consent withdrawal, please see our Privacy Policy

Reviewer #1: No

Reviewer #2: No

---

## [Author Response · Author response to Decision Letter 2]

29 Jan 2026

Please see the attachment labeled "Response to Reviewers (26Jan26)" for a summary of the most recent reviewer feedback we received and an itemized list of our responses.

---

## [Decision Letter · Decision Letter 2]

5 Feb 2026

Risk and protective factors for incidents of intimate partner violence among active-duty military personnel

PONE-D-25-34092R2

Dear Dr. Stander,

We’re pleased to inform you that your manuscript has been judged scientifically suitable for publication and will be formally accepted for publication once it meets all outstanding technical requirements.

Kind regards,

Yordanis Enríquez Canto, Ph.D.

Academic Editor

PLOS One

Additional Editor Comments (optional):

Reviewers' comments:

Reviewer's Responses to Questions

**Comments to the Author**

Reviewer #1: All comments have been addressed

Reviewer #2: All comments have been addressed

2. Is the manuscript technically sound, and do the data support the conclusions?

Reviewer #1: (No Response)

Reviewer #2: (No Response)

3. Has the statistical analysis been performed appropriately and rigorously?

Reviewer #1: (No Response)

Reviewer #2: (No Response)

4. Have the authors made all data underlying the findings in their manuscript fully available?

Reviewer #1: (No Response)

Reviewer #2: (No Response)

5. Is the manuscript presented in an intelligible fashion and written in standard English?

Reviewer #1: (No Response)

Reviewer #2: (No Response)

Reviewer #1: (No Response)

Reviewer #2: (No Response)

**Do you want your identity to be public for this peer review?** For information about this choice, including consent withdrawal, please see our Privacy Policy

Reviewer #1: No

Reviewer #2: No

---

## [Editor Report · Acceptance letter]

PONE-D-25-34092R2

PLOS One

Dear Dr. Stander,

I'm pleased to inform you that your manuscript has been deemed suitable for publication in PLOS One. Congratulations! Your manuscript is now being handed over to our production team.

Kind regards,

on behalf of

Prof. Yordanis Enríquez Canto

Academic Editor

PLOS One